# Comparative Effectiveness of Various Eradication Regimens for Helicobacter Pylori Infection in the Northeastern Region of Poland

**DOI:** 10.3390/ijerph19116921

**Published:** 2022-06-06

**Authors:** Justyna Wasielica-Berger, Patryk Gugnacki, Maryla Mlynarczyk, Pawel Rogalski, Agnieszka Swidnicka-Siergiejko, Stefania Antonowicz, Michalina Krzyzak, Dominik Maslach, Andrzej Dabrowski, Jaroslaw Daniluk

**Affiliations:** 1Department of Gastroenterology and Internal Medicine, Medical University of Bialystok, 15-276 Bialystok, Poland; justyna.wasielica-berger@umb.edu.pl (J.W.-B.); progalsky@gmail.com (P.R.); agnieszka.swidnicka-siergiejko@umb.edu.pl (A.S.-S.); antonowiczstefania@gmail.com (S.A.); adabrows@umb.edu.pl (A.D.); 2Department of Oncology, Medical University of Bialystok, 15-025 Bialystok, Poland; patryk.gugnacki@umb.edu.pl; 3Department of Histology and Cytophysiology, Medical University of Bialystok, 15-025 Bialystok, Poland; maryla.mlynarczyk@umb.edu.pl; 4Department of Hygiene, Epidemiology and Ergonomics, Medical University of Bialystok, 15-022 Bialystok, Poland; michalina.krzyzak@umb.edu.pl; 5Department of Public Health, Medical University of Bialystok, 15-295 Bialystok, Poland; dominikm@umb.edu.pl

**Keywords:** bismuth-containing quadruple therapy, eradication, *Helicobacter pylori*, levofloxacin-based triple therapy, metronidazole-based triple therapy, Poland

## Abstract

Purpose: Due to the lack of systematic data on antibiotic sensitivity, the treatment of the highly prevalent and pathogenic *Helicobacter pylori* (*H. pylori*) infection still poses a significant problem. Therefore, the aim of our study was to compare the efficacy of the three most commonly used anti-*H. pylori* therapies in northeastern Poland. Patients and Methods: This was a retrospective, single-center study performed on 289 outpatients with an *H. pylori* infection. Patients received one of the following three treatment regimens: (1) bismuth quadruple therapy (BQT) for 10 days, (2) metronidazole-based triple therapy (M-TT) for 10 or 14 days, and (3) levofloxacin-based triple therapy (L-TT) for 10 or 14 days. Results: BQT, M-TT, and L-TT accounted for 93.2% of prescribed anti-*H. pylori* therapies. The overall success rate for all treatment regimens was 84.1% (243/289). The effectiveness of first- and second-line therapy was similar and reached 83.8% and 86.2%, respectively. The efficacy of the individual treatment regimens was as follows: (1) BQT—89.4% (84/94), (2) M-TT—80.6% (112/139) and 78.8% (26/33) for 10 and 14 days, respectively, and (3) L-TT—84.6% (11/13) and 100% (10/10) for 10 and 14 days, respectively. The overall duration of treatment and type and dose of proton pump inhibitor (PPI) had no effect on the treatment efficacy. Conclusions: In the northeastern part of Poland, 10-day BQT and 10- or 14-day L-TT are effective treatment regimens for *H. pylori* eradication and have appear to be superior to M-TT. Practitioners in our clinic followed mostly local anti-*H. pylori* therapy guidelines.

## 1. Introduction

*Helicobacter pylori* (*H. pylori*) is one of the most common bacterial infections, affecting 4.4 billion people worldwide, with large interregional variations in incidence ranging from 19% in Switzerland to 88% in Nigeria [1]. In the meta-analysis evaluating data from 73 countries and 6 continents, the prevalence of *H. pylori* infection was 44.3% (95% CI: 40.9–47.7) worldwide and as much as 66.9% in Poland [2]. Moreover, the frequency of infection increases with age and affects 32% of Polish children and 84% of adults [3]. Although asymptomatic in a part of the infected individuals, *H. pylori* can be responsible for the development of severe symptoms and complications. Most common consequences of the infection are gastritis, dyspepsia, and peptic ulcer disease. The latter carries a 3.5% mortality rate and is the reason for nearly 20,000 hospitalizations each year in the US, making it a significant economic burden to the healthcare system [4]. Longstanding *H. pylori* infection is a very strong risk factor for gastric cancer or mucosa-associated lymphoid tissue (MALT) lymphoma [5]. The risk of progression of precancerous lesions in the stomach is higher in infected individuals and increases with the duration of infection [6]. It is estimated that *H. pylori* was responsible for 810,000 new cases of non-cardia adenocarcinoma in 2018 worldwide [7]. Despite the advances in diagnostic and treatment modalities, mortality from gastric cancer is still high, reaching 74% [8].

According to the Maastricht V/Florence Consensus Report, *H. pylori* is considered an infectious disease, and therefore treatment should be offered to all infected patients, independent of symptoms [9]. Treatment of this infection has been shown to be beneficial in reducing the incidence of peptic ulcer disease and gastric cancer [10,11,12]. Several meta-analyses have shown that the eradication of *H. pylori* decreased the risk of gastric cancer from approximately 2 to nearly 5 times [11,12,13], and one even proved that eradication reduces mortality from gastric cancer [12].

Therefore, the key issue is to choose an effective eradication treatment that will allow for the optimal elimination of the infection in the first attempt. Due to the changes in the sensitivity of bacteria to antibiotics, the problem of anti-*H. pylori* therapy is still current and should be monitored. In recent years, a 7-fold decline in the efficacy of standard triple therapy has been observed in patients with clarithromycin-resistant *H. pylori* infections [14]. In addition, over a ten-year period (2006–2016), resistance to the other two antibiotics used in the triple therapy, namely metronidazole and levofloxacin, exceeded the 15% threshold in most World Health Organization (WHO) regions [14]. This is due to the increasing and uncontrolled use of these antibiotics to treat both *H. pylori* and other systemic infections (64% increased global fluoroquinolone usage in the time period 2000–2010). The use of levofloxacin in patients with *H. pylori* resistant to this antibiotic increases the risk of treatment failure by 8-fold, while in the case of resistance to metronidazole, the risk increases only 2.5-fold [14]. Importantly, metronidazole resistance can be partially overcome by increasing the dose and duration of treatment, especially in combination with bismuth therapy. Data from a meta-analysis showed that the resistance rate for clarithromycin, metronidazole, and levofloxacin in Europe was 18%, 32%, and 11%, respectively [14]. In Poland, resistance to clarithromycin and metronidazole is even higher and reaches 46% and 56%, respectively [15]. The current guidelines from the European *Helicobacter* and Microbiota Study Group (Maastricht V/Florence Consensus Report) and the American College of Gastroenterology (ACG) recommend bismuth-based quadruple therapy (BQT) or concomitant non-bismuth quadruple therapy (when bismuth is not available) in areas of high clarithromycin resistance (>15%) for the first-line treatment, instead of a standard clarithromycin-based regimen [9,16,17]. The preferred duration of treatment is 14 days; however, a 10-day treatment is acceptable if found to be effective locally. Local Polish guidelines also allow for the use of metronidazole-based triple therapy (M-TT) or sequential quadruple therapy as a first-line treatment [18]. Due to the low availability of culture and susceptibility tests, the therapy should be based in clinical practice on the patient’s earlier exposure to antibiotics and the local occurrence of resistance to the most commonly used drugs. However, local data on the efficacy of eradication depending on treatment regimen and duration are not available in most cases. In the recently published European Registry on *Helicobacter pylori* management, only 69 patients (0.3% of total examined population) were enrolled in Poland [19]. Therefore, the aim of our study was to compare the efficacy of 10-day BQT and M-TT or levofloxacin-based triple therapy (L-TT) prescribed for 10 or 14 days for the treatment of *H. pylori* infections in northeastern Poland.

## 2. Materials and Methods

### 2.1. Patients

This was a retrospective, single-center study conducted on outpatients from the Department of Gastroenterology and Internal Medicine at the Medical University of Bialystok, which is the largest gastroenterology clinic in northeastern Poland. We collected the medical records of adult patients treated for *H. pylori* infection from January 2017 to December 2020. The indications for *H. pylori* testing were as follows: (1) dyspepsia, (2) peptic ulcer disease, (3) sideropenic anemia of unknown reason, (4) family history of gastric cancer, and (5) MALT lymphoma. As the Maastricht V/Florence Consensus Report recommends antibiotic susceptibility testing prior to third-line treatment for *H. pylori* infection, we only included patients with first or second eradication attempts. The collected information included demographic (age, sex) and clinical data (tests used to detect *H. pylori*, indications for eradication, treatment scheme and duration). Patients with insufficient clinical data (unknown treatment regimen and duration) or treatment discontinuation (described as consumption of less than 90% of prescribed drugs) were excluded from further analysis.

### 2.2. Tests Used to Confirm H. pylori Infection

*H. pylori* infection was confirmed by non-invasive (stool antigen, serum *H. pylori* antibodies) or invasive tests (upper GI endoscopic examination with rapid urease test and/or histopathology). The decision about the test selection was made individually for each patient by the attending physician who followed the routine, general rules. If the patient had clinical indications for endoscopy, an invasive method was chosen. In the case of no other indication for endoscopy than for *H. pylori* diagnostics or if the patient did not agree for endoscopy, the non-invasive method was selected. The selection of the type of test during endoscopy was up to the performing endoscopist.

The *H. pylori* stool antigen test (SAT) was determined by qualitative immunochromatographic assay (Certest Biotec S.L., Zaragoza, Spain). The Certest is a validated test with high sensitivity (94%) and specificity (98%) for the detection of *H. pylori*. Chemiluminescent immunoassay technology (The LIAISON^®^ H. pylori IgG, DiaSorin, Stillwater, MN, USA) was used for the qualitative determination of immunoglobulin G (IgG) antibodies to *H. pylori* in human serum. The principal components of the test are magnetic particles coated with an *H. pylori* antigen and anti-human IgG monoclonal antibodies labelled with an isoluminol derivative. The light signal was measured by a photomultiplier, as relative light units and values ≥ 0.9 were considered positive.

For the urease rapid test (URT, Lencomm Trade Intl., Warsaw, Poland), two biopsy samples from the antrum and gastric body were placed in a urea broth that contained the pH indicator phenol red. Assessments were made after 5 and 60 min. For the histopathologic diagnosis of *H. pylori* infection, two biopsies from the antrum and two from the gastric body were collected according to the Maastricht V guidelines [9]. The samples were fixed with 10% PBS-buffered formalin, embedded in paraffin, stained with hematoxylin and eosin (H&E), and examined in light microscopy (Olympus BX45) by an experienced gastrointestinal pathologist. In cases of chronic active gastritis in which *H. pylori* was not detected after H&E staining, ancillary Giemsa staining was executed [9,20].

### 2.3. Treatment Regimens for Eradication

We have searched through the records of 319 patients that received treatment for *H. pylori* in our clinic. Out of all prescribed anti-*H. pylori* treatment regimens, we have chosen the three most frequent for further analysis. The three most frequent regimens were: bismuth quadruple therapy (BQT) for 10 days, metronidazole-based triple therapy (M-TT) for 10 or 14 days, or levofloxacin-based triple therapy (L-TT) prescribed for 10 or 14 days. 

BQT consisted of a proton pump inhibitor (PPI) b.i.d. and 3 capsules containing bismuth subcitrate potassium (140 mg) + metronidazole (125 mg) + tetracycline hydrochloride (125 mg) (Pylera^®^, Allergan Pharmaceuticals International Limited Clonshaugh Business & Technology Park, Dublin, Ireland) q.i.d. We chose a drug containing a combination of bismuth, metronidazole, and tetracycline in one capsule because it facilitates its use by patients and improves compliance. The doses of antibacterial drugs in this preparation were set by the manufacturer and could not be modified. The effectiveness of the drug was confirmed in the current guidelines [9]. M-TT included PPI b.i.d., amoxicillin (1000 mg) b.i.d., and metronidazole (500 mg) b.i.d. L-TT consisted of PPI b.i.d., levofloxacin (250 mg) b.i.d., and amoxicillin (1000 mg) b.i.d. The dosing of amoxicillin, levofloxacin, and metronidazole was based on current European and ACG guidelines [9,16]. The following PPIs were used in eradication therapy: esomeprazole (40 mg), omeprazole (20 mg), pantoprazole (40 mg), and lansoprazole (30 mg). All patients were instructed to discontinue PPI at least 2 weeks prior to an endoscopic examination or stool antigen test.

The endpoint of the study was the eradication rate after each treatment regimen. Successful eradication was confirmed by a stool antigen test, rapid urease test, or a histological evaluation performed 4 weeks or more after the completion of treatment (ranging 4 weeks to 9 months). The selection of the treatment regimen and the method of *H. pylori* eradication control depended on the attending physician. 

### 2.4. Ethics of the Study

This investigation was reviewed and approved by the Local Ethic Committee (approval number: R-I-002/133/2018). Due to the retrospective study design, no written informed consent was obtained from the study participants. However, all patients who underwent endoscopy signed a standard written informed consent for upper GI endoscopy with biopsy before examination.

### 2.5. Statistical Analysis

The analysis of the *H. pylori* infections was run in statistical software R, version 4.1.0 (R Core Team (2021), R: Language and environment for statistical computing by R Foundation for Statistical Computing, Vienna, Austria). The description of the nominal variables was based on the number of observations and % structure. Age, as the only numeric variable, was described with basic descriptive statistics and visualized with a histogram. In order to assess the dependency of treatment effectiveness on the given factors, appropriate statistical tests were used (Pearson’s chi-square or Fisher exact test). The tests verified whether the difference in the proportion of each factor between successful and failed treatment was statistically significant. The significance level assumed was *p* = 0.05. The factors analyzed were: type of treatment (antibiotic), IPP, duration of treatment, and eradication. Additionally, the analysis of effectiveness dependency on several factors (type of treatment, type or dose of IPP, and duration of treatment) was repeated by splitting the data into first-line and second-line eradication cases.

## 3. Results

Of the 319 patients initially recruited, 9 were excluded due to lack of sufficient clinical data. The BQT, M-TT, and L-TT together accounted for 93.2% of all anti-*H. pylori* therapies (289 out of 310 therapies), leading to 289 patients being included in the analysis. The baseline characteristics of the patients, as well as the diagnostic and treatment methods used in the study, are presented in Figure 1 and Table 1.

More than half of the participants were female (n = 187, 64.7%). The average age was 60.93 (SD = 13.56) years, ranging from 20 to 87 years. The main indication for anti-*H. pylori* therapy was dyspepsia (73%), peptic ulcer disease (12.5%), family history of gastric cancer (8.3%), sideropenic anemia (5,5%), and MALT lymphoma (0,7%). *H. pylori* infection was diagnosed by URT (47%), SAT (27%), histopathology (19%), and serum *H. pylori* IgG antibodies (7%). 

The vast majority of patients (90%, 260/289) were treated for the first time. The most common first-line treatment regimen was M-TT, which was administered to 169 of 260 patients (65%), followed by BQT (79/260, 30.4%) and L-TT (12/260, 4.6%). For the second-line therapy (after prior eradication failure), BQT was the most often prescribed (15/29), followed by L-TT (11/29) and M-TT (3/29). All the patients in BQT were treated for 10 days (32.5%), while in M-TT more participants received the treatment for 10 days (48.1%) than for 14 days (11.4%). In the L-TT group, patients received treatment for 10 days (4.5%) or 14 days (3.5%). Esomeprazole was the most commonly prescribed PPI (44.3%). Subsequently, omeprazole (36.3%), pantoprazole (17.6%), and lansoprazole (1.7%) were used. There were no differences between the study groups in the baseline demographic data, indications for anti-*H. pylori* therapy, and type of PPI used.

The overall eradication rate for all treatment regimens was 84.1% (243/289), while in 15.9% (46/289) of patients, the treatment failed to eradicate *H. pylori*. The effectiveness of treatment ranged from 76.5% to 100.0% depending on the type of antibiotic regimen, IPP, treatment duration, and treatment attempt (Figure 2).

The effectiveness of the most commonly used M-TT was 80.6% (112/139) and 78.8% (26/33) for 10- and 14-day treatment durations, respectively. BQT therapy, which was the second most common treatment scheme used in our study, had a better eradication rate (89.4%, 84/94) than M-TT, but the difference was not statistically significant. L-TT administered for 10 and 14 days was successful in 84.6% (11/13) and 100% (10/10) of cases, respectively. However, due to the small number of patients treated with L-TT, the results once again were not statistically significant. The duration of treatment, regardless of the selected schedule, had no effect on the treatment efficacy and was 84.1% for both 10-day and 14-day treatments. None of the prescribed PPIs had an advantage in increasing the effectiveness of the therapy. The effectiveness of first- and second-line therapy was similar and reached 83.8% and 86.2%, respectively. Summarizing the obtained results, none of the analyzed factors differentiated by a statistically significant effectiveness rate (Table 2).

The influence of the selected factors on the treatment effectiveness was also determined with a split into all first-line treatments and all second-line treatments. As a result, no factor differentiating the effectiveness of the treatment in a statistically significant way within each of the eradication groups was found (*p* > 0.05 in all cases) (Table 3).

## 4. Discussion

*H. pylori* eradication is crucial in treating gastritis, peptic ulcer disease, and preventing gastric adenocarcinoma. Unfortunately, the resistance of *H. pylori* to antibiotics is high and the treatment is not well tolerated by some patients. Therefore, it is important to monitor the efficacy of particular treatment regiments, especially as this efficacy may vary locally depending on the population studied.

Our study revealed that BQT prescribed for 10 days as well as L-TT for 10 or 14 days are effective for *H. pylori* treatment and have an advantage over M-TT in the area of high clarithromycin resistance; however, the difference was not statistically significant. Contrary to other bacterial infections, e.g., urinary tract infections, where usually a culture test with antibiotic sensitivity is available, the treatment of *H. pylori* is most often empirical, with only little data on the local antibiotic resistance profile of the bacteria. Guidelines for the eradication of *H. pylori* suggest using antibiotic susceptibility data whenever possible. However, in most cases, this data is not available. As the number of antibiotics used in the eradication of *H. pylori* is limited (clarithromycin, amoxicillin, metronidazole, levofloxacin, tetracycline, rifabutin), increasing antibiotic resistance is a significant clinical problem. Therefore, it is very important to develop local reports on treatment efficacy to assist the clinician in selecting the optimal treatment for a particular population based on current data. Empirical therapy based on the accurate medical history of prior antibiotic exposure has been shown to be as effective as genotypic resistance-guided therapy in a resistant *H. pylori* infection [21].

It is believed that optimal therapy against *H. pylori* should be at least 90% effective [22]. Unfortunately, such values are rarely found in clinical practice and the problem of treatment failure is encountered very often. Our overall eradication efficacy was 83.8% and 86.2% for the first-line and second-line treatment, respectively, which is clearly below the recommended 90% eradication rate. However, these data represent real clinical practice and are very similar to the results obtained from the European *H. pylori* Management Register (Hp-EuReg). Based on an analysis of 21,533 infected individuals from 27 countries, the overall modified intention-to-treat (mITT) efficacy of empirical first-line treatment was 85.6% [19]. The most often prescribed in this study were triple therapies (48.8%), with clarithromycin-containing triple therapies in first place (44.3%). When the data were limited to the southeastern region of Europe only (the region that included Poland), the frequency of triple therapies was as high as 81.1%, with clarithromycin-containing therapies in 78.3% of cases [19]. Interestingly, the frequency of prescribing M-TT among all regimens in Hp-EuReg was only 2.6% in Europe as a whole and 2.1% in southeastern Europe. In our study, M-TT was the most commonly prescribed eradication regimen with 59.5% (172/289) of patients. This may be explained by the high compliance with the local, Polish guidelines that propose M-TT therapy on par with BQT, non-bismuth quadruple therapy, and sequence quadruple therapy in first-line treatment. The other reason for the frequent choice of this therapy was the relatively low cost. 

For many years, the standard of treatment was triple therapy with PPI and two antibiotics out of clarithromycin, metronidazole, and amoxicillin for 7–10 days with a high success rate of up to 90% [23]. Unfortunately, the success of triple therapy has declined sharply in recent years due to increasing resistance to clarithromycin, which reached 30% in Italy, for example [24]. In Poland, resistance to clarithromycin ranges from 26% to 46% [15,25]. Therefore, for our local standards for *H. pylori* eradication therapy, clarithromycin has been replaced with metronidazole. The resistance rate to metronidazole in Poland is also high, but it has less impact on treatment due to the synergistic effect of metronidazole with other drugs and the lack of a direct transition between in vitro and in vivo results [26]. Our study found that 10-day M-TT achieved an 80.6% treatment success. These results are consistent with routine clinical practice data obtained in the European Registry on *H. pylori* management, where the effectiveness of M-TT in Europe treated as a whole was 84.2% and 80% in the region of southeastern Europe, where Poland was accounted [19]. In relation to individual European countries, the results of M-TT in our study were similar to those obtained in Italy (80%), but higher than in Russia (77%) or Spain (69%) [27]. To summarize, both our local data and those obtained from the European registry clearly show that the efficacy of M-TT is suboptimal and therefore should be abandoned in some areas, including the northeastern part of Poland. Fortunately, based on the available data, the prescription rate of triple therapy has decreased from more than 50% in 2013 to 32% in 2018 [19]. 

BQT, the recommended first-line treatment by major guidelines, was the second most common eradication regimen in our study—32.5% (94/289). This is more often than the average in Europe. The data from the European registry show that in 2013–2018, the frequency of BQT use accounted for 15.7% of all prescribed eradication regimens. Use of bismuth quadruple therapies increased from 0–2% in 2013/2014 to 20% in 2018 [19]. The reason for the high frequency of BQT prescription in our study is the fact that it covered the years 2017–2020, i.e., the period when the use of bismuth increased significantly in accordance with the guidelines. BQT is highly effective in treating *H. pylori*, exceeding the 90% eradication efficacy, even in areas with high clarithromycin resistance [28,29,30,31]. Similarly, in our study, the effectiveness of BQT was high, reaching 89.4%, which is very close to the recommended 90% threshold. Our results are consistent with data from other European countries. The effectiveness of the 10-day BQT was 87% in Russia, 90% in Spain, and 93% in Italy [27]. These results clearly indicate that BQT is an effective therapeutic option in the treatment of H. pylori infection not only in northeastern Poland, but also in other regions of Europe. 

The third eradication regimen used in our study was L-TT. The Maastricht V/Florence Consensus Report did not consider levofloxacin-based therapy as a first-line treatment option [9]. However, the authors of the guidelines added that L-TT can be used in an area of high dual resistance to clarithromycin and metronidazole and low resistance to levofloxacin. ACG guidelines list levofloxacin therapy as an acceptable first-line option [16]. At the same time, it is emphasized that the resistance to levofloxacin is increasing, reaching 31% in the United States [32]. Data on *H. pylori* resistance to levofloxacin in Poland are very limited. The molecular analysis of *H. pylori* strains in southwestern Poland showed a low, 6% resistance rate to levofloxacin [15]. In our study, first-line therapy with levofloxacin was used in only 11 (3.8%) patients and was effective among 10 of them. A recently published network meta-analysis of 22,975 patients randomized to 8 first-line regimens has found that, in Western countries, the best eradication rate was achieved by L-TT (88.5%) [33]. These results are in line with our data and may indicate a similar pattern of levofloxacin resistance between these regions. However, this data differs from the results of Hp-EuReg (European countries only included), where L-TT had a poor 46.3% effectiveness [19]. The differences may be due to the greater resistance to levofloxacin in some regions of Europe compared to Poland. Taking into account the continuous increase in resistance to levofloxacin and the potential side effects of quinolones, L-TT should be used primarily as a rescue treatment.

The current guidelines of the major gastroenterological organizations (Toronto Consensus, Maastricht V/Florence Consensus, ACG) agree that second-line therapy should be based on bismuth quadruple therapy or levofloxacin triple therapy, depending on what was used previously [9,16,17]. Our study found that BQT and L-TT used in a second-line therapy had a high acceptable efficacy of 93.3% and 90.9%, respectively, which is in line with recommendations. However, the number of patients receiving the second-line treatment was small (15 and 11 patients, respectively). M-TT as a second-line therapy was used in only three patients, but it was ineffective in two of them, which clearly indicates that such a treatment regimen should not be used in Poland. The BQT was more efficient as a second-line than as a first-line treatment (93.3% vs. 88.6%, difference not significant). It is possible that after one failed therapy, the doctors tried to better explain the rationale, potential difficulties, and side effects that might occur during the treatment, thus achieving a more motivated approach and better adherence of patients to the treatment schedule [34].

According to a recent study, longer treatment duration was associated with higher eradication rates [19]. However, an analysis of the European registry showed that most patients were eradicated within 7–10 days (69%), while only 31% of them were prescribed 14 days of therapy. Toronto Consensus recommends 14-day durations for all treatment regimens [17]. European and American guidelines allow a 10-day BQT if its effectiveness has been proven locally [9,16]. Importantly, our study found that BQT used for 10 days was very effective, reaching almost 90% eradication success. These results are slightly lower than in the Hp-EuReg study, where 10 days of BQT treatment achieved an eradication rate of 94.6% [19] but may still be a promising method of *H. pylori* treatment in the northeastern part of Poland. Data from other European countries confirmed that 10-day BQT is as effective as a 14-day therapy (87% versus 86% in Russia and 90% versus 89% in Spain, respectively) [27]. Two randomized control trials performed in Europe and North America tested triple-capsule-containing bismuth, metronidazole, and tetracycline plus omeprazole for ten days and reported intention-to-treat eradication rates of 80% and 87.7%, respectively [35,36]. Another important finding of our study is the fact that extending the M-TT to 14 days did not increase the eradication rate compared to the 10-day regimen (78.8% vs. 80.6%, respectively). This data indicates that M-TT should not be recommended in Poland, regardless of the duration of treatment. At the same time, extending the L-TT to 14 days resulted in a 100% effectiveness of eradication treatment; however, due to the small number of patients, it is hard to draw clear conclusions.

It has been shown that the use of higher PPI doses in triple therapy with clarithromycin increased the eradication efficacy by 11% [37]. Therefore, Maastricht V/Florence Consensus suggests the use of higher PPI doses in eradication therapy [9]. In our study, we did not show that the type of PPI had an impact on the effectiveness of eradication, although we used esomeprazole in a double standard dose. The potential explanation for this observation is the fact that we have not used clarithromycin therapy wherein a beneficial effect of higher gastric acid inhibition was demonstrated.

The efficacy of treatment may also depend on the type of *H. pylori* strain. At least two meta-analyses have shown that the chance of eradication is 8–11% higher in patients infected with cagA (+) strains [38,39]. The eradication rate is also approximately 10% greater in strains possessing allele s1 compared to s2 of the vac A gene [39]. The identification of cagA (−) or vacA s2 strains may be an indication of more aggressive treatment regimens. We did not perform strain typing in the present study, but the data from a previous multicenter study conducted, among others, on the population of our region of Poland showed that cagA (+) strains accounted for 64.1% and vacA s1 for 66.4% of all strains [40].

Supplementary therapy with specific probiotics is aimed at increasing the effectiveness of *H. pylori* eradication and reducing treatment-related side effects. The European guidelines regard certain probiotics (*Saccharomyces boulardii*, *Lactobacillus*, *Bacillus clausii*), which should be considered in order to reduce side effects in individual cases (e.g., susceptible patients with comorbidities or poor antibiotic tolerance) [9]. However, data from several meta-analyses on the use of one or multiple strains of probiotics showed little additional beneficial effects on eradication [26]. Therefore, the Toronto and the ACG guidelines do not recommend their use [16,17]. In our daily clinical practice, due to the low quality of evidence on the efficacy of probiotics, they have not been used in the treatment of patients infected with *H. pylori*. More research is definitely needed in this area.

Our study has some limitations. Most notably, the study was conducted retrospectively, so we were unable to obtain more information about patients (i.e., smoking, socioeconomic status, adverse effects of treatment). Secondly, we used various tests to detect *H. pylori* infection or confirm its eradication. However, this represents the real-time data from daily practice. Moreover, all tests have been validated by our certified laboratory or experienced pathologists. The advantages of the study are a big study group that received uniform, modern treatment in accordance with international and Polish recommendations regarding both diagnostics and treatment. The single center model of the study limits the influence of various environmental factors on the results.

## 5. Conclusions

In conclusion, 10-day BQT and 10- or 14-day L-TT are effective treatment regimens for *H. pylori* eradication and tend to be superior to M-TT among the population of northeastern Poland, although the differences were not significant. Only BQT and L-TT approached the expected 90% effectiveness threshold.

## Figures and Tables

**Figure 1 ijerph-19-06921-f001:**
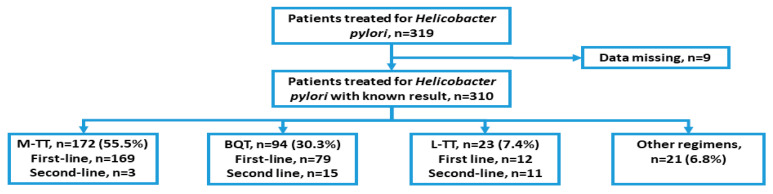
Flow chart of the overall study design and treatment regimens for the first-line and second-line therapy. M-TT—metronidazole-based triple therapy, BQT—bismuth quadruple therapy, L-TT—levofloxacin-based triple therapy.

**Figure 2 ijerph-19-06921-f002:**
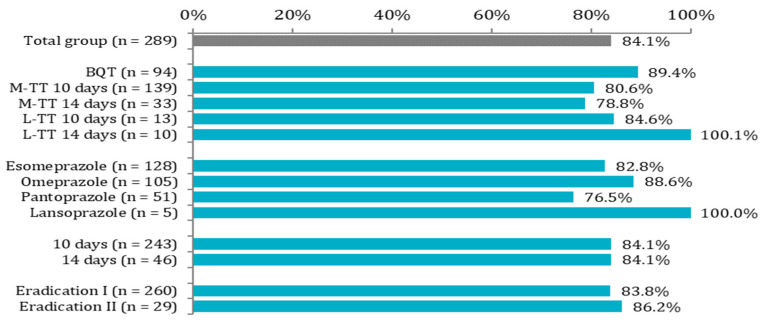
Comparison of the effectiveness rate in the total group and between treatment regimens, type of PPI, treatment duration, and eradication attempts. BQT—bismuth-based quadruple therapy, M-TT—metronidazole-based triple therapy, L-TT—levofloxacin-based triple therapy, PPI—proton pump inhibitor.

**Table 1 ijerph-19-06921-t001:** Baseline characteristic, diagnostic, and treatment methods of patients included in the analysis.

		Number of Patients	Percentage of Patients
Sex	Female	187	64.7%
	Male	102	35.3%
Indication to test *H. pylori* infection	Dyspepsia	211	73%
	Peptic ulcer disease	36	12.5%
	Family history of gastric cancer	24	8.3%
	Anaemia	16	5.5%
	MALT lymphoma	2	0.7%
Test to detect *H. pylori* infection	Urease rapid test	135	47%
	Stool antigen test	78	27%
	Histopathology	54	19%
	*H. pylori* IgG antibody	22	7%
Treatment regimen	Bismuth quadruple therapy	94	32.5%
	Metronidazole-based triple therapy		
	10 days	139	48.1%
	14 days	33	11.4%
	Levofloxacin-based triple therapy		
	10 days	13	4.5%
	14 days	10	3.5%
Proton pump inhibitor	Esomeprazole	128	44.3%
	Omeprazole	105	36.3%
	Pantoprazole	51	17.6%
	Lansoprazole	5	1.7%
*H. pylori* eradication attempt	First-line therapy	260	90.0%
	Second-line therapy	29	10.0%

**Table 2 ijerph-19-06921-t002:** Treatment effectiveness depending on the treatment regimen, type of IPP, treatment duration, and eradication attempts.

Variable		Successful		Unsuccessful		*p*
		n	%	n	%	
Treatment regimen	Bismuth quadruple therapy	84	34.6%	10	21.7%	0.213
	Metronidazole-based triple therapy 10 days	112	46.1%	27	58.7%	
	Metronidazole-based triple therapy 14 days	26	10.7%	7	15.2%	
	Levofloxacin-based triple therapy 10 days	11	4.5%	2	4.3%	
	Levofloxacin-based triple therapy 14 days	10	4.1%	0	0.0%	
Proton pump inhibitor	Esomeprazole	106	43.6%	22	47.8%	0.180
	Omeprazole	93	38.3%	12	26.1%	
	Pantoprazole	39	16.0%	12	26.1%	
	Lansoprazole	5	2.1%	0	0.0%	
*H. pylori* eradication attempt	First-line therapy	218	89.7%	42	91.3%	0.951
	Second-line therapy	25	10.3%	4	8.7%	

Proportion analyses were performed with chi-square Pearson tests.

**Table 3 ijerph-19-06921-t003:** Comparison of the treatment effectiveness depending on the treatment type, IPP, and treatment duration with a split into first- and second-line therapy.

Variables		First-Line Therapy (n = 260)			Second-Line Therapy (n = 29)		
		Successful (n)	Unsuccessful (n)	*p*	Successful (n)	Unsuccessful (n)	*p*
Treatment regimen	BQT	70 (88.6%)	9 (11.4%)	0.243	14 (93.3%)	1 (6.7%)	0.077 ^1^
	M-TT	136 (80.5%)	33 (19.5%)		1 (33.3%)	2 (66.7%)	
	L-TT	11 (91.7%)	1 (8.3%)		10 (90.9%)	1 (9.1%)	
Proton pump inhibitor	Esomeprazole	88 (83.0%)	18 (17.0%)	0.222 ^1^	18 (81.8%)	4 (18.2%)	>0.999 ^1^
	Omeprazole	89 (88.1%)	12 (11.9%)		4 (100%)	0 (0.0%)	
	Pantoprazole	37 (75.5%)	12 (24.5%)		2 (100%)	0 (0.0%)	
	Lansoprazole	4 (100%)	0 (0.0%)		1 (100%)	0 (0.0%)	
Tratment duration	10	185 (83.7)	36 (16.3%)	>0.999	21 (87.5%)	3 (12.5%)	0.557 ^1^
	14	33 (84.6%)	6 (15.4%)		4 (80.0%)	1 (20.0%)	

Proportion analyses were performed with chi-square Pearson tests or Fisher exact test ^1^. BQT—Bismuth quadruple therapy; M-TT–Metronidazole-based triple therapy; L-TT—Levofloxacin-based triple therapy.

## Data Availability

The data supporting reported results can be obtained from the corresponding author on request.

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
