# Peer review of "Comparative Effectiveness of Various Eradication Regimens for Helicobacter Pylori Infection in the Northeastern Region of Poland"

_ijerph, 2022, doi:10.3390/ijerph19116921_

Round 1

Reviewer 1 Report

In this manuscript, the author Wasielica-Berger et al. describe the Comparative effectiveness of various eradication regimens for Helicobacter pylori infection in the north-eastern region of Poland. In this study, they compare the efficacy of three treatment regimens; bismuth quadruple therapy (BQT) prescribed for 10 days; metronidazole-based triple therapy (M-TT) for 10 or 14 days, and levofloxacin-based triple therapy (L-TT) for 10 or 14 days in the treatment of H. pylori infections in north-eastern Poland. The study is appropriately conducted. The parameters in the study are properly evaluated and the manuscript is well written. Methods and results are clearly discussed in the manuscript. The experimental data reported are convincing and the impact of the study is clearly beneficial. While this information, in general, is of interest and add to the current literature illustrating the effectiveness of various eradication regimens, there are several points that have to be addressed by the authors and I have some suggestions regarding the presentation and interpretation of the data.

Major critics:

1.    Do the authors confirm the type of H. pylori strain that exists in this particular region of the country and hence the treatment regimen had been selected for this study? It would be great to emphasize and discuss the knowledge of genomics that could help in identifying the virulent strains and their selective treatment regimen. Targeting such virulent strains and developing aggressive eradication strategies aimed specifically at population subgroups may represent the path forward.  

2.    Does probiotics supplementatio006E exert any favorable effect on H. pylori, and does it has any significance to reduce the incidence of side effects related to antimicrobial therapy? If yes, is it not worth studying the treatment regimen that includes the use of probiotics.

3.    Authors did explain the treatment regimen in the materials and methods section but were unable to provide the details regarding the selection of particular doses of the treatment. E.g. muth subcitrate potassium (140mg), metronidazole (125mg), tetracycline hydrochloride 138 (125mg) etc. It would be great either provide the evidence for the dose selection criteria or quote the reference to support these treatment regimens.

Minor critiques:

1.    The study design and experiments are appropriate, and it is advised that the authors should provide a schematic /flow chart of the overall study design and treatment regimens for readers to quickly grasp the study.

2.    Several definite and indefinite articles missing along with some grammatical errors, also please correct the manuscript for undefined spaces.

3.    It is advised that the authors should define abbreviations at least in the first instance, please check the manuscript thoroughly e.g. MALT lymphoma (page 2, line 52).

Author Response

Thank you for your thoughtful and thorough review of our manuscript and for your constructive comments. Here are the replies to the issues you have raised.

MAJOR CRITIQUES

  1. Do the authors confirm the type of H. pylori strain that exists in this particular region of the country and hence the treatment regimen had been selected for this study? It would be great to emphasize and discuss the knowledge of genomics that could help in identifying the virulent strains and their selective treatment regimen. Targeting such virulent strains and developing aggressive eradication strategies aimed specifically at population subgroups may represent the path forward.

Answer: Indeed, the presence of the cagA (+) or vacAs1 strains increase the chances for successful H. pylori eradication. Identification of cagA (-) or vacAs2 strains may be an indication of more aggressive treatment regimens. Unfortunately, genotyping the strain of H. pylori is not a routine practice in Poland. As this was an observational retrospective study, we were not able to tell exactly what strains infected the currently studied patients. However, our Department previously took part in a multicenter study titled “Genotypic and clinical differences of seropositive Helicobacter pylori children and adults in the polish population” and two authors of the present study (Justyna Wasielica-Berger and Jaroslaw Daniluk) were the members of the team appointed to carry out that study (Task Force of the Polish Society of Gastroenterology). The data collected during this previous study showed that the prevalence of cagA (+) strains in adult polish population was 64.1% and 66.4% of strains had s1 allel of vacA gen.

To address your comment, we made changes in the manuscript. Line: 380-387

  1. Does probiotics supplementation exert any favorable effect on H. pylori, and does it has any significance to reduce the incidence of side effects related to antimicrobial therapy? If yes, is it not worth studying the treatment regimen that includes the use of probiotics.

Answer: Thank you for this comment. Probiotics may counteract the harmful effects of antibiotics on gut microbiota. However, the data on the effectiveness of this approach are inconclusive. The current guidelines also do not provide sufficient evidence for the use of probiotics in the eradication of H. pylori. Therefore, in our study, patients were not prescribed probiotics during eradication. We have added a paragraph in the Discussion section explaining the role of a probiotic in the treatment of H. pylori infection. Line: 388-397

  1. Authors did explain the treatment regimen in the materials and methods section but were unable to provide the details regarding the selection of particular doses of the treatment. E.g. bismuth subcitrate potassium (140mg), metronidazole (125mg), tetracycline hydrochloride (125mg) etc. It would be great either provide the evidence for the dose selection criteria or quote the reference to support these treatment regimens

Answer: We chose a drug containing a combination of bismuth, metronidazole, and tetracycline in one capsule, because it facilitates its use by patients and improves compliance. The doses of antibacterial drugs in this preparation were set by the manufacturer and could not be modified. The effectiveness of the drug was confirmed in the current guidelines. The dosing of amoxicillin, levofloxacin and metronidazole was based on current European and ACG guidelines.

The above explanations have been included in section 2.3 Treatment regimens for eradication. Line: 150-157.

MINOR CRITIQUES

  1. The study design and experiments are appropriate, and it is advised that the authors should provide a schematic /flow chart of the overall study design and treatment regimens for readers to quickly grasp the study.

Answer: We have provided a flow chart of the overall study design and treatment regimens (Figure 1). Line: 190-194

  1. Several definite and indefinite articles missing along with some grammatical errors, also please correct the manuscript for undefined spaces.

Answer: The article has been re-checked and corrected for grammatical errors, missing articles and undefined spaces.

  1. It is advised that the authors should define abbreviations at least in the first instance, please check the manuscript thoroughly e.g. MALT lymphoma (page 2, line 52).

Answer: Thank you for this comment. The abbreviations were defined: Line 34; 53; 128

Reviewer 2 Report

In the paper by Rokkas et. al. (DOI:https://doi.org/10.1053/j.gastro.2021.04.012), the levo-triple therapy was shown to achieve the highest eradication rates in Western countries. How does the data from these meta-analysis data compare with the findings in the current manuscript?

On a separate note, in recent years, increased resistance to metronidazole and levofloxacin has resulted in them also becoming less effective when used empirically in triple therapies. I would love to see some insights into this from the authors in the present manuscript. 

Author Response

Thank you for your very valuable comments. Here are the replies to the issues you have raised.

  1. In the paper by Rokkas et. al. (DOI:https://doi.org/10.1053/j.gastro.2021.04.012), the levo-triple therapy was shown to achieve the highest eradication rates in Western countries. How does the data from these meta-analysis data compare with the findings in the current manuscript?

Answer: Thank you for this very interesting suggestion. Rokkas et al. compared the efficacy of all empirical first-line regimens versus standard triple treatment using a network meta-analysis of published randomized controlled trials. Based on analysis of 22,975 patients randomized to 8 first line regimens, they found that only vonoprazan triple therapy and reverse hybrid therapy achieved cure rates of >90%. However, in Western countries, the best eradication rate was achieved by L-TT (88.5%). These results are in line with our data and may indicate a similar pattern of levofloxacin resistance between these regions. However, the group of patients treated with L-TT in our study was too small to draw any definitive conclusions.

To address your comment, we made changes in the manuscript. Line: 330-339

  1. On a separate note, in recent years, increased resistance to metronidazole and levofloxacin has resulted in them also becoming less effective when used empirically in triple therapies. I would love to see some insights into this from the authors in the present manuscript.

Answer: Thank you for drawing our attention to this very important issue. The increased resistance to metronidazole and levofloxacin has resulted in significant decrease in the usage of triple regimens in many areas of the world. The use of levofloxacin in patients with H. pylori resistant to this antibiotic increases the risk of treatment failure by 8-fold, while in the case of resistance to metronidazole, the risk increases 2.5-fold. Importantly, metronidazole resistance can be partially overcome by increasing the dose and duration of treatment, especially in combination with bismuth therapy.

To respond to your comment, we have included data from Savoldi et al. study (Reference 14) reporting alarming rates of antibiotic resistance worldwide. Line: 70-80

Reviewer 3 Report

The study is very well written and easy to read. The study was well designed, informative, it has sufficient power for statistical analysis, the analyses are appropriate, the graphics are appropriate and conclusions are supported by data. 

If there is one recommendation to be made, is that the authors extend the discussion of how their findings compare with data available from other Western, Central and Eastern European countries.

Author Response

We thank you for this positive and encouraging comment. Here are the reply to the issue You have raised.

  1. The study is very well written and easy to read. The study was well designed, informative, it has sufficient power for statistical analysis, the analyses are appropriate, the graphics are appropriate and conclusions are supported by data.

If there is one recommendation to be made, is that the authors extend the discussion of how their findings compare with data available from other Western, Central and Eastern European countries.

Answer: Thank you for this valuable comment. Indeed, the effectiveness of eradication may vary across Europe. We did not find data for all European countries, but based on the European H. pylori Management Register (Hp-EuReg), we included data from Spain, Italy, and Russia which enrolled the largest number of patients treated for H. pylori infection. Therefore, we compared the M-TT (Line: 299-301), and BQT (Line: 316-320) results from our study with those from the countries mentioned. Data comparing the effectiveness of 10-day and 14-day BQT in Russia and Spain were also included (Line:362-364).

Reviewer 4 Report

The manuscript ijerph-1734135 entitled: “Comparative effectiveness of various eradication regimens for Helicobacter pylori infection in the north-eastern region of Poland” by Justyna Wasielica-Berger and coworkers aimed to compare the efficacy of three most commonly used anti-H pylori therapies in north-eastern Poland: 1) bismuth quadruple therapy (BQT) for 10 days, 2) metronidazole-based 26 triple therapy (M-TT) for 10 or 14 days, 3) levofloxacin-based triple therapy (L-TT) for 10 or 14 days.

The overall success rate for all treatment regimens was 84.1% (243/289). The effectiveness of first- and second-line therapy was similar and reached 83.8% and 86.2%, respectively. The efficacy of the individual treatment regimens was as follows: 1) BQT - 89.4% (84/94), 2) M-TT - 80.6% (112/139) and 78.8% (26/33) for 10 and 14 days, respectively, 3) L-TT - 84.6% (11/13) and 100% (10/10) for 10 and 14 days, respectively. The overall duration of treatment, type and dose of PPI had no effect on the treatment efficacy. In the north-eastern part of Poland, 10-day BQT and 10- or 14-day L-TT are effective treatment regimens for H. pylori eradication and have a trend to be superior to M-TT.

Despite is a single center study, the design is clear, the methodology used is consistent of adequate.

Conclusions are consistent with the results.

Minor comments: table 1 and table 3 are splitted in 2 different pages

Author Response

We thank you for this positive and encouraging comment.

  1. Despite is a single center study, the design is clear, the methodology used is consistent of adequate. Conclusions are consistent with the results. Minor comments: table 1 and table 3 are splitted in 2 different pages

Answer: We improved the layout of the manuscript so that Tables 2 and 3 would not be split into separate pages